# Regulation of Leaderless mRNA Translation in Bacteria

**DOI:** 10.3390/microorganisms10040723

**Published:** 2022-03-28

**Authors:** Lorenzo Eugenio Leiva, Assaf Katz

**Affiliations:** 1Programa de Biología Celular y Molecular, ICBM, Facultad de Medicina, Universidad de Chile, Santiago 8380453, Chile; lleiva@chapman.edu; 2Facultad de Ciencias, Universidad de Chile, Santiago 7800003, Chile

**Keywords:** leaderless mRNA, lmRNA, bacterial translation, expression control, translation regulation

## Abstract

In bacteria, the translation of genetic information can begin through at least three different mechanisms: canonical or Shine-Dalgarno-led initiation, readthrough or 70S scanning initiation, or leaderless initiation. Here, we discuss the main features and regulation of the last, which is characterized mainly by the ability of 70S ribosomal particles to bind to AUG located at or near the 5′ end of mRNAs to initiate translation. These leaderless mRNAs (lmRNAs) are rare in enterobacteria, such as *Escherichia coli*, but are common in other bacteria, such as *Mycobacterium tuberculosis* and *Deinococcus deserti,* where they may represent more than 20% and even up to 60% of the genes. Given that lmRNAs are devoid of a 5′ untranslated region and the Shine-Dalgarno sequence located within it, the mechanism of translation regulation must depend on molecular strategies that are different from what has been observed in the Shine-Dalgarno-led translation. Diverse regulatory mechanisms have been proposed, including the processing of ribosomal RNA and changes in the abundance of translation factors, but all of them produce global changes in the initiation of lmRNA translation. Thus, further research will be required to understand how the initiation of the translation of particular lmRNA genes is regulated.

## 1. Introduction

The process of translation is of utmost importance to life, as it allows the production of proteins based on the genetic information contained in RNA. Translation can be divided into several steps: usually, a step where the initiation site is recognized by the ribosome and polymerization begins (initiation), a step where amino acids carried by tRNAs are sequentially added to the nascent peptide (elongation), and a step where polymerization stops, releasing the nascent peptide (termination). Translation elongation and termination proceed by similar mechanisms in prokaryotes and eukaryotes. For instance, in all forms of life, the recognition of amino acids in the ribosome depends on two steps: the correct aminoacylation of tRNAs by aminoacyl-tRNA synthetases, followed by proper pairing between three nucleotides in the mRNA (codon) and three nucleotides in a tRNA (anticodon) that carry amino acids [1,2,3]. In contrast to this conservation, the identification of translation initiation sites varies. For instance, the canonical initiation of translation in eukaryotes requires the recognition of 5′ capping by the eukaryotic initiation factor 4F complex (together with additional signals such an initiation codon) [2]. On the other hand, canonical initiation in bacteria is completely independent of the chemical characteristics of the 5′ end of mRNA and commonly begins at sites that are far from this 5′ extreme, allowing the organization of genes in operons [1]. Furthermore, within both eukaryotes and prokaryotes, there are alternative mechanisms that allow the initiation of translation at other, noncanonical, sites [2,4].

In this review, we focus on one such mechanism, leaderless mRNA (lmRNA) translation, by which bacterial ribosomes recognize an initiation codon that is very near to the 5′ end of an mRNA to begin translation. It must be noted that eukaryotes also present non-canonical initiation mechanisms, explanations of which can be found elsewhere [2]. Nevertheless, our focus is mainly on lmRNA translation in *Escherichia coli*, which has been an important model for studying the mechanisms and regulation of bacterial lmRNA translation. Other bacteria are also considered, although this is limited because there is far less information available on them.

## 2. Mechanisms for Bacterial Initiation of Translation

Before turning our attention to the translation of lmRNAs, we will briefly introduce the canonical or Shine-Dalgarno (SD)-led initiation of translation (Figure 1, left), which will be used as a reference throughout this text. In bacteria, translation usually begins with the formation of a complex between the ribosome binding site (RBS) on the mRNA, the small subunit of the ribosome (30S), and the initiator tRNA charged with *N*-formylmethionine (fMet-tRNA^fMet^). The RBS is recognized through interactions between a sequence located at the ribosomal 3′ end (anti-SD (aSD) sequence) and a complementary sequence located upstream of the coding sequence (SD sequence). Moreover, the binding of the initial codon to the anticodon of the initiator tRNA and an unfolded region where the 30S particle can bind mRNA are of utmost importance to allow the correct recognition of the RBS [5,6,7]. In addition to the role played by the aSD sequence and the initiator tRNA, the bS1 protein (previously S1 [8]) assists in canonical translation. In *E. coli*, this large ribosomal protein of the small subunit is composed of six “S1” domains, each one presenting an OB fold that is usually associated with RNA binding. In fact, most of these domains in bS1 bind to RNA and participate in the unfolding of structured RNA. This unfolding is usually required to increase the accessibility of the SD and initiation codon, essential for SD-led translation. Additionally, the RNA-binding domains of bS1 assist in mRNA recognition by binding enhancer sequences located upstream of the SD. Consequentially, bS1 is essential for canonical translation in proteobacteria, particularly when the 5′ UTR is strongly structured. However, in other groups, such as low G/C content Gram-positive bacteria (e.g., *Bacillus*), bS1 is absent from the ribosomes [9,10,11,12].

Finally, in addition to the role of the 5′ UTR in RBS recognition, sequences downstream of the starting codon may also affect ribosome binding [13]. After the formation of a 30S-mRNA complex, the large subunit (50S) can bind to form a complete ribosome (70S) at the RBS, allowing translation to begin with the entrance of the first elongator aa-tRNA and peptide bond formation. The entire process is accompanied and enhanced by initiation factors (IF) 1, 2, and 3, which ensure that translation initiates only if a canonical RBS and initiator tRNA have bound to the 30S. For example, IF2 recruits the fMet-tRNA^fMet^ initiator tRNA, while IF3 enhances the selection of initiator over elongator tRNAs, interferes with the large subunit’s association, and helps to discriminate against sites that differ from the canonical RBS. IF1 enhances the activity of the two other initiation factors. The dissociation or displacement of all these factors is required to allow the binding of the 50S subunit and formation of a complete ribosomal particle [1].

In addition to the canonical mechanism of translation initiation, bacteria can initiate translation through at least two other mechanisms [1]. One is 70S scanning, also called translational coupling or readthrough (Figure 1, center). In this mechanism, the ribosome does not dissociate to 30S and 50S subunits after finishing the translation of a cistron. Instead, it continues to be associated with the mRNA and slides downstream until it reaches a canonical RBS where it can initiate translation. In addition to allowing initiation of genes downstream of canonical genes, 70S scanning allows the initiation of genes located at the beginning of a transcript, as long as the 5′ UTR is unstructured and there are 70S particles available for initiation. Similar to canonical initiation, IF3 is required for this process. By contrast, IF1 is not essential, although it is strongly stimulatory [4,14].

Finally, translation can also initiate from lmRNAs (Figure 1, right), which are mRNAs that lack a canonical RBS. In this case, the initiation codon is at or very near to the 5′ end of the mRNA. Thus, these RNAs are practically devoid of the leader or 5′ untranslated region (5′ UTR) and completely lack an SD or other sequence from the 5′ UTR that guides 30S binding for the canonical initiation of translation. In effect, in the presence of IF3, 30S does not bind to model lmRNAs in vitro [19]. Instead, lmRNA translation begins mostly by the direct binding of a 70S particle to an initiation codon that is at or very proximal to the 5′ end of an mRNA [20,21,22]. In general, translation initiation factors seem to have a less relevant role in lmRNA translation than that observed for SD-led translation [20,23]. Perhaps this is because lmRNAs bind directly to 70S particles and do not require IFs to modulate 70S formation. Nevertheless, as discussed below, IFs do alter the behavior of lmRNA translation and are necessary for the translation of at least some lmRNAs.

There is limited information about the features that help with recognition of the initiation codon in lmRNA by ribosomes. It is known that, in *E. coli*, lmRNA translation starts at the first AUG site. Alternative codons such as GUG, UUG, and CUG are much less efficient, with some not showing any protein production at all, although this loss of translation efficiency is stronger in artificial [24,25] than natural [26] lmRNAs. A similar trend is observed in some species, such as *Caulobacter crescentus* [27], while in others, initiation from non-AUG codons is more efficient; for instance, GUG in *Mycobacterium smegmatis* [25], *Saccharopolyspora erythraea* [28], and *Streptomyces coelicolor* [29]. How these initiation codons are recognized is still not known. In SD-led translation, interaction between the initiation codon and initiator tRNA is essential for the recognition of the initiation site. Consequently, an SD-led mRNA where the initiation codon has been replaced by an amber stop codon (UAG) will not allow translation initiation. Nevertheless, if an amber-suppressing initiator tRNA is present, canonical translation can proceed. In contrast to canonical translation, lmRNA translation cannot initiate in a similar setup, even if an amber-suppressing initiator tRNA is present. Thus, the selection of an AUG codon does not depend solely on binding by the initiator tRNA, and the mechanism behind its recognition selectivity is currently not understood [24].

In addition to the initiation codon, other characteristics are also expected to assist in lmRNA recognition by the translation machinery, as the loss of translation efficiency is smaller in natural than in synthetic lmRNAs when the initiation codon is changed from AUG to GUG (see above). For instance, it has been shown that the presence of a phosphate at the 5′ end is required for lmRNA translation [30], and CA repeats downstream and near the start codon are known to strongly enhance translation [13]. By contrast, the distance from the 5′ end to the start codon [27,31] and the stability of the structures surrounding this codon [27] have been observed to strongly decrease lmRNA translation in some bacterial species. Although these observations might partially explain the important variations that have been observed in efficiency of translation between diverse lmRNAs [20,22,25], further research will be required to fully understand the mechanisms of lmRNA selection by the ribosome in *E. coli* and, particularly, in other organisms (Figure 2).

## 3. Regulation of lmRNA Translation

Although there are some contradictory results [5,34], initiation is considered the slowest step of translation for most genes and, thus, the step where translation regulation is most frequently observed [5,35]. In fact, canonical initiation may be regulated through diverse strategies at this stage. For instance, there are RNAs that alter their structure in response to changes in the concentrations of some metabolites. The modification of such structures, called riboswitches, may alter the accessibility of the RBS for 30S, consequentially modifying the efficiency of translation initiation. The binding of small RNAs or proteins to or near the RBS may also alter its accessibility for 30S producing similar effects on translation efficiency [35,36].

Although the mechanisms involved in the regulation of canonical initiation are well described, we do not have a good description of lmRNA translation regulation. Moll et al. proposed that, in *E. coli*, lmRNA translation could be regulated by the cleavage the 16S rRNA. In their model, under stress conditions, the MazF toxin was activated due to decreased transcription and/or translation, coupled to the faster degradation of its antitoxin, MazE. As MazE was rapidly degraded, the remaining MazF led to the cleavage of 16S rRNA. As a consequence, the ribosome lost the aSD sequence (or at least its functionality [37]) and, with it, the possibility of binding canonical mRNAs to initiate translation [38]. It was proposed that, as a consequence, mostly lmRNAs would be translated under such conditions. Additionally, some lmRNAs could be formed by a similar MazF-dependent cleavage of canonical mRNAs. Thus, MazF activation would lead to the formation of lmRNAs and “stress ribosomes” that would specifically translate such mRNAs [38]. Unfortunately, other laboratories have not been able to reproduce these findings, either by MazF over-expression [39,40] or under natural stress conditions that enhance lmRNA translation [41]. Consequently, we must assume that the MazF dependent regulation of lmRNA translation is not as widespread as originally proposed.

Before the MazF model for the regulation of lmRNA translation was proposed, published results pointed in a different direction, as in vitro and some in vivo experiments suggested that alterations to the concentration or activity of diverse components of the translation apparatus could enhance lmRNA translation to the detriment of canonical translation (Figure 2 and Table 1). These early results can be summarized as follows:

1.-bS1 and other ribosomal proteins, such as uS2 (previously S2 [8]), are not required for the translation of lmRNAs [11,20,42,43]. Although there is a small decrease in in vitro translation with the loss of these proteins, increased translation is observed in vivo when they are lost (e.g., because of a temperature-sensitive mutation of the uS2 coding gene, *rpsB*). This is probably a consequence of the loss of the ability to translate canonical mRNAs that, under “normal” conditions, would compete for available ribosomes [43].2.-In vitro results indicate that translation initiation factors are not essential for lmRNA translation, but the process is usually stimulated by their presence. While IF1 and IF2 stimulate lmRNA translation in vitro at any concentration, IF3 has a dual effect. Low concentrations enhance lmRNA translation, but an increased concentration of this factor inhibits protein production from some lmRNAs [19,20]. Supporting this effect, in vivo experiments have shown that increased expression of *infC* (which codes for IF3) inhibits the translation of model lmRNAs and using *infC* mutants that allow greater flexibility in initiation codon selection enhance their expression [19]. It has been proposed that these results are a consequence of the role of IF3 in discriminating against noncanonical RBS [19] or derive from its ability to enhance 70S dissociation [19,20]. In any case, the effects of initiation factors seem to depend, to some extent, on the lmRNA sequence. For instance, although Udagawa et al. observed increased lmRNA translation when IF2 was used [20], O’Donnell and Janssen observed that IF2 inhibited the association of 70S with some, but not all, lmRNAs [21]. Additionally, in contrast to reports showing that IF3 may not be required for lmRNA translation [19,20], Yamamoto et al. showed that this is only observed when high levels of fMet-tRNA^fMet^ and mRNA are present, while, with “normal” concentrations, lmRNA translation requires IF3 [4] (although, as discussed in point 3, others have suggested that this would also depend on the lmRNA sequence [20]).3.-Increased availability of initiator tRNA (fMet-tRNA^fMet^) strongly enhances lmRNA translation, while producing only small increases in canonical translation. Under increased fMet-tRNA^fMet^ concentrations, some lmRNAs, such as *cI* from λ phage, stop requiring initiation factors for translation, while other tested synthetic lmRNAs still require the presence of IFs for efficient translation [20].

Recent publications have proposed additional possibilities for the modulation of lmRNA translation. Landwehr et al. proposed that YchF ATPase from *E. coli* inhibits lmRNA translation [44], and we found that (p) ppGpp may activate lmRNAs translation in addition to its known role in the inhibition of canonical translation [41]. Whether these and the previously reported effects are interconnected or not is not known, but several associations may exist. For instance, the (p) ppGpp inhibition of canonical translation may lead to the accumulation of fMet-tRNA^fMet^, which could stimulate lmRNA translation. Its binding to initiator factors [45,46] could have a similar effect by decreasing the competition for ribosomes between canonical and lmRNA translation. A connection between YchF and initiation factors is also possible, and, in fact, Landwehr et al. proposed that this protein regulates lmRNA production owing to its interaction with IF3 [44].

Unfortunately, although we know that alterations in the levels of components or regulators of the translation machinery alter the efficiency of lmRNA translation in *E. coli*, there is little information on the role of these phenomena in the physiological regulation of translation. Moll et al. observed increased cleavage of 16S rRNA in nutrient-poor media [38], and we observed increased (p) ppGpp-dependent translation of lmRNA under oxidative stress [41], but to our knowledge, there are no reports on the effects of other conditions that may alter the concentrations or chemical modifications of components of the translation machinery. For example, the concentration of IFs in *E. coli* is known to increase under cold shock [47,48], and these proteins as well as bS1 are phosphorylated during infections by T7 phage [49,50,51]. As discussed previously, these changes might alter lmRNA translation, but we are unaware of any studies on lmRNA translation in these conditions. Such reports will be essential for determining the relevance of these phenomena in the physiological regulation of lmRNA translation.

## 4. Regulation of the Expression of Specific Genes by lmRNA Translation

Most of the reported mechanisms that could alter the efficiency of lmRNA translation are rather general, affecting all lmRNAs at the same time. Although it has been observed that, for many of these effects, there is some impact of the sequence on the intensity of lmRNA translation enhancement, an important question remains as to whether lmRNA translation allows for the regulation of the expression of particular genes or if only global changes to gene translation are possible. In enterobacteria such as *E. coli*, only a small number of lmRNAs have been found to be expressed under control conditions. For instance, one deep sequencing study found only 20 to 30 lmRNAs in the *E. coli* transcriptome [52]. Thus, increased global lmRNA translation combined with the activation of a subgroup of promoters that enhance lmRNA transcription could be enough to elicit specific responses. For instance, in *Shigella flexneri*, a promoter located within the *virF* coding sequence has been found to control the transcription of a shorter version of VirF (Virf_21_). VirF is a transcription factor that controls the pathogenicity of *Shigella*, and VirF_21_ has been found to repress the transcription of the long version of *virF* [53]. Thus, increased transcription of *virF_21_* coupled with enhanced translation of all lmRNAs could potentially modulate pathogenicity, even if lmRNA translation activation is general and not specific. 

In contrast to enterobacteria, in which only a small number of lmRNAs have been observed, in other bacteria, a much greater fraction of the transcriptome corresponds to lmRNAs. For example, transcriptomic studies suggest that in *Mycobacterium smegmatis* [25], *Mycobacterium tuberculosis* [54], and *Streptomyces coelicolor* [55], around 20 to 25% of the genes depend on lmRNA translation, while in organisms such as *Deinococcus deserti* [56], this number increases to around 60% of the genes. It has been proposed that this important variation in the number of lmRNAs found in diverse species is inversely correlated with the number and length of operons in the corresponding genomes [57]. In species with large numbers of lmRNAs, one would expect that there would be a mechanism to specifically regulate the translation of some genes, as global changes would otherwise alter the translation of too many genes to allow specific adaptation to each possible environmental change.

Given that lmRNA presents very few, if any, nucleotides upstream of the initiation codon, it is unlikely that we will find riboswitches controlling lmRNA translation, unless they are located within the coding sequence. Nevertheless, there is no obvious reason to think that we will not find proteins or small RNAs that bind to lmRNAs, altering the accessibility of these RNAs for the translation apparatus. It is also plausible that the chemical modification of the 5′ ends of lmRNAs through bacterial capping with NADH [58], Np4 [59,60,61], or other molecules [62] might alter translation efficiency. Although, to our knowledge, there are no reports of such regulation strategies, we do know that lmRNA translation could specifically regulate the translation of downstream genes. Beck et al. proposed that small leaderless open reading frames can regulate the translation of downstream genes using a strategy that is similar to uORFs in eukaryotic cells. In fact, they showed that 5′ “untranslated” regions in *E. coli* beginning with AUG can alter the translation efficiency for the genes immediately downstream [6,22,63]. Furthermore, Canestrari et al. showed that similar small lmRNA ORFs found in *M. smegmatis* and containing contiguous cysteine codons could regulate the translation of downstream genes through an attenuation mechanism in which the speed of ribosome elongation depends on the availability of cysteine [64]. Thus, although we lack enough information to make strong conclusions, it is reasonable to expect that further research will show diverse strategies for the regulation of the translation of specific lmRNA genes, at least in organisms in which such a mechanism is used for the translation of an important fraction of the transcriptome.

## Figures and Tables

**Figure 1 microorganisms-10-00723-f001:**
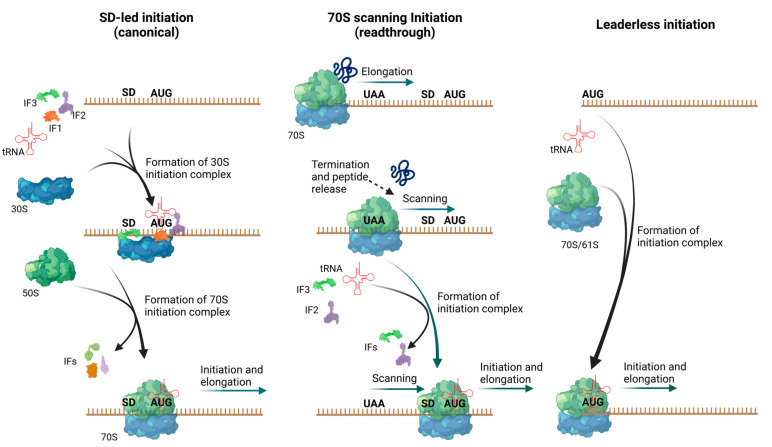
Mechanisms of translation initiation in bacteria. In bacteria, translation can be initiated by three mechanisms. Shine-Dalgarno (SD)-led translation (**left**), 70S scanning (**center**), and leaderless initiation (**right**). As shown, although most of the components are similar between these three mechanisms, translation initiation factors (IFs) have different roles. Arrows indicate either movement on mRNA or binding/unbinding of translation machinery components. Figure created with BioRender.com. Model of IF1 is based on pdb 1AH9 [15] and model of IF2 on pdb 4b48 [16]; model of IF3 was obtained from AlphaFold (P0A707) [17,18].

**Figure 2 microorganisms-10-00723-f002:**
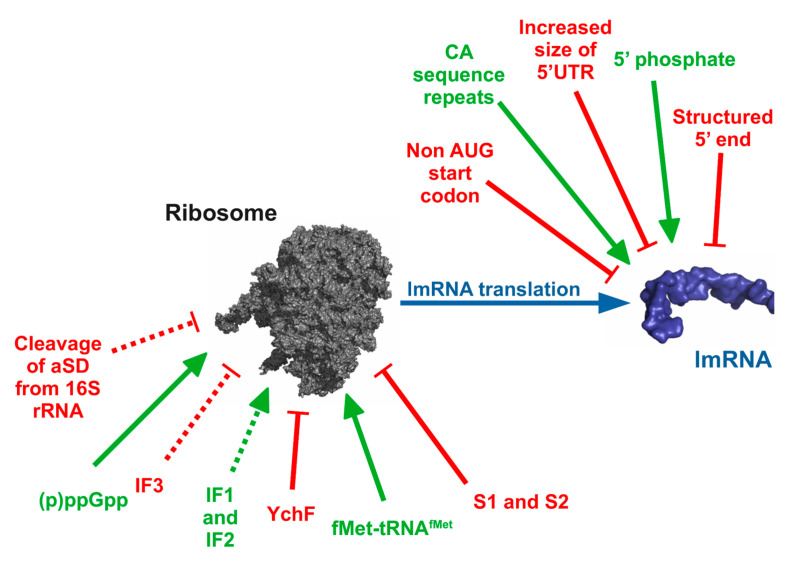
Control of lmRNA translation. It has been observed that several factors or leaderless mRNA (lmRNA) properties may alter lmRNA translation efficiency. Shown at the top are lmRNA properties that enhance (green) or repress (red) lmRNA translation, and at the bottom are other factors that alter lmRNA translation, probably through interaction with ribosome. Dotted lines indicate factors where changing conditions or strains produce discordant results. Please refer to main text for full details regarding each case. Models of ribosomal and RNA structures were constructed using PYMOL molecular graphics system [32] and pdb 4V9M [33].

**Table 1 microorganisms-10-00723-t001:** Comparison between SD-led (canonical) and leaderless translation.

	SD Led Initiation	Leaderless Initiation	References
5′ UTR	Essential for ribosome binding	Inhibits ribosome binding	[1,5,27,31]
Structures around initiation codon	Inhibit initiation	Inhibit initiation	[1,5,27]
CA repeats downstream of initiation codon	Enhance translation	Enhance translation	[13]
aSD	Required for translation	Dispensable, at least in some strains	[38,39,40]
5′ phosphate	No effect	Enhances translation	[30]
S1 and S2	Required for translation	Mildly inhibit translation	[20,42,43]
IF1 and IF2	Required for translation	Enhance translation	[19,20]
IF3	Required for translation	Inhibits translation	[19,20]
YchF	Not required for translation	Inhibits translation	[44]
fMet-tRNA^fMet^	Mildly enhances translation	Strongly enhances translation	[20]
(p)ppGpp	Inhibits translation	Activates translation	[41]
sRNA	Might stimulate or inhibit translation	Unknown	[36]
Riboswitches	Might stimulate or inhibit translation	Unknown	[36]
Proteins binding around RBS	Might stimulate or inhibit translation	Unknown	[36]

## Data Availability

Not applicable.

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
