# Peer review of "Regulation of Leaderless mRNA Translation in Bacteria"

_microorganisms, 2022, doi:10.3390/microorganisms10040723_

Round 1
Reviewer 1 Report
The review “Translation regulation of leaderless mRNA in bacteria” by Lorenzo Eugenio Leiva and Assaf Katz focuses on the mechanisms of translation initiation of leaderless mRNAs (lmRNAs) in bacteria and discusses their possible regulatory mechanisms. Alternative molecular strategies leading to the translation of lmRNAs lacking 5′ untranslated region and Shine-Dalgarno sequence are compared to canonical translation mechanisms.
The introduction is not up to the standard of an article dealing with the regulation of mRNA translation. The first lines (26-41) focus, in oversimplified language, on the chemical properties of DNA and describe mRNA as “another nucleic acid that is mostly a copy of DNA”. A presentation of the key players in mRNA translation initiation and the importance of alternative mechanisms of translation initiation for the regulation of gene expression would be more appropriate.
A Figure showing current models of lmRNA translation initiation pathways should be included.
The pivotal role of ribosomal protein S1 in canonical bacterial translation could be described in more details (line 69). This would help the authors to better emphasize that, in contrast, S1 is dispensable for lmRNA translation (line 202).
Paragraph 4 could be reorganized to better highlight the levels of differential lmRNA regulation.1) translational efficiency of lmRNA can be modulated depending on the availability of components of the translational machinery or 2) under stress conditions.
Figure 1 is devoted to the regulation of lmRNA translation but is not cited in the text. The structures of a ribosome and lmRNA (?) are shown in Figure 1 but neither the source of these data nor the origin of these structures are indicated. The legend of Figure 1 contains truncated sentences. This should be corrected. This figure should be improved overall: A schematic of the typical regulatory elements and sequences at the 5’ end of lmRNA should be included and would be more informative.
Reviewer 2 Report
Manuscript ID: microorganisms-1614587
Title: Regulation of Leaderless mRNA Translation in Bacteria
Leiva et al. provide a detailed summary on the translation regulation of leaderless mRNAs (lmRNAs). While their review covers the relevant literature concerning the model organism E.coli and other prokaryotic species, the manuscript, as it stands, appears premature for the following reasons.
Major points:
In general, I got the impression that this review is lacking a clear direction. It appears like an enumeration of facts. In line with this, I´m missing conceptual clarity and a more stringent intellectual discourse. I feel the review would benefit a lot if the authors would have a stronger focus on the regulation of Leaderless mRNA Translation in Bacteria as suggested by the title.
The thoughts about the origin of live, and how lmRNAs could be related to it, are interesting but maybe too speculative and too farfetched for a review focusing on the regulation of lmRNA based gene expression.
Conversely, the introduction starts at a quite basal level, explaining genetic principles most of which supposed to be common knowledge for the potential reader of this review.
However, more than anything else, the review is desperately in need of attractive illustrations. The review contains only one Figure that is neither very informative, nor well integrated in the text section. Actually, the text in its current form does not refer to the Figure at all. I suggest using more than one display item (if possible), and a table or a flow chart to better illustrate the most important concepts, principles and potential mechanisms discussed in this review.
Minor points:
- line 69: The authors should refer to the now commonly used nomenclature (Ban et al., Current
Opinion in Structural Biology, 2014): bS1, uS2 etc.
- line 86: The term gene should be replaced by mRNA or cistron.
- line 136: What exactly do the authors mean with protein driven catalysis?
- line 141: “In fact, as discussed in the previous paragraphs, in contrast to what is observed in
canonical mRNAs, translation initiation sites in lmRNAs are simple, are located very near to the 5’
end and are much less dependent on initiation factors that is observed in canonical mRNAs.” The
meaning of this sentence is not clear.
-line 265: “In such scenarios, one would expect there should be a mechanism to specifically regulate the translation of some genes, as global changes would otherwise alter translation of too many genes to allow specific adaptation to each possible environmental change.” Is it known if and to which extent regulation if transcription contributes to the regulation of lmRNA translation?
Round 2
Reviewer 1 Report
The review “Translation regulation of leaderless mRNA in bacteria” by Lorenzo Eugenio Leiva and Assaf Katz has been significantly improved by changes in the introduction, several corrections and paragraph reorganizations.
The addition of a new figure summarizing the different mechanisms of translation initiation found in bacteria and of a new Table comparing canonical and leaderless translation initiation increase the clarity of the manuscript. These improvements make the review more informative.
Minor: Part of Legend 1 has been truncated.
Author Response
Following the recommendations of both reviewers we have sent the manuscript to a professional English correction service. These changes are marked in the document using the word processor changes tracking system. In addition to the corrections suggested by the English corrector, we have made the following modifications to answer the comments of the reviewers:
figure 1: a tRNA was added to the initiation complex of lmRNA translation (right row). Also, an explanation was added to the arrows that didn’t have it. In the previous version, there was a formatting error in the word file that hid the caption behind the figure. We apologize for this inconvenience. Now the caption should be visible.
Previous line 57: “comparison point” was changed to “reference”
Previous line 117: “other”changed to “some”
Previous lines 120-122: “In contrast to canonical translation, lmRNA translation cannot initiate in lmRNA starting with an amber (UAG) stop codon when an amber-suppressing initiator tRNA is present.” was replaced by “In SD led translation, interaction between the initiation codon and initiator tRNA is essential for the recognition of the initiation site. Consequently, a SD led mRNA where the initiation codon has been replaced by an amber stop codon (UAG) will not allow translation initiation. Nevertheless, if an amber-suppressing initiator tRNA is present, canonical translation can proceed. In contrast to canonical translation, lmRNA translation cannot initiate in a similar setup, even if an amber-suppressing initiator tRNA is present”
Previous line 160: replicate was changed to reproduce
Previous line 304: Ifs was replaced by IFs
The use of subjunctives was strongly reduced
Reviewer 2 Report
Following the reviewer´s suggestions, Leiva et al. improved their manuscript by reorganizing the text and adding a figure and a table.
While the manuscript benefits from the changes conceptually, unfortunately a number of linguistic mistakes remained or appeared.
For instance, the authors tend to make extensive use of the subjunctive (lines 94, 97, 150, 153, 156, 158, 159, 179, 190, 191 etc.). I´ve got the impression that in more than 90% of the cases, it is either not required (and unnecessarily weakens the statements), or even out of place. The authors should fix this and other language problems, utilizing a professional (English) language editing service.
Some case examples are pointed out here. However, this list for sure is not complete:
- line 57: reference or benchmark instead of comparison point
- line 117: some instead of other
- lines 120-122: “In contrast to canonical translation, lmRNA translation
cannot initiate in lmRNA starting with an amber (UAG) stop codon when an
amber-suppressing initiator tRNA is present.”
The meaning of this sentence in the given context is not clear to me.
- line 160: reproduce instead of replicate?
- line 304: IFs instead of Ifs
Figures and table:
- The captions of Fig. 1 and Table 1 are not present!
- Fig. 1 leaderless initiation third row: Where is the tRNA? Shouldn´t there be
a tRNA at the start codon?
Author Response

(The authors gave the same response as above.)
